# The Role of Osteoprotegerin and Its Ligands in Vascular Function

**DOI:** 10.3390/ijms20030705

**Published:** 2019-02-06

**Authors:** Luc Rochette, Alexandre Meloux, Eve Rigal, Marianne Zeller, Yves Cottin, Catherine Vergely

**Affiliations:** 1Equipe d’Accueil (EA 7460): Physiopathologie et Epidémiologie Cérébro-Cardiovasculaires (PEC2), Université de Bourgogne–Franche Comté, Faculté des Sciences de Santé, 7 Bd Jeanne d’Arc, 21000 Dijon, France; alexandre.meloux@gmail.com (A.M.); eve.rigal@u-bourgogne.fr (E.R.); marianne.zeller@u-bourgogne.fr (M.Z.); yves.cottin@chu-dijon.fr (Y.C.); catherine.vergely@u-bourgogne.fr (C.V.); 2Service de Cardiologie–CHU-Dijon, 21000 Dijon, France

**Keywords:** osteoprotegerin, OPG/RANKL/RANK, endothelium, vascular disease

## Abstract

The superfamily of tumor necrosis factor (TNF) receptors includes osteoprotegerin (OPG) and its ligands, which are receptor activators of nuclear factor kappa-B ligand (RANKL) and TNF-related apoptosis-inducing ligand (TRAIL). The OPG/RANKL/RANK system plays an active role in pathological angiogenesis and inflammation as well as cell survival. It has been demonstrated that there is crosstalk between endothelial cells and osteoblasts during osteogenesis, thus establishing a connection between angiogenesis and osteogenesis. This OPG/RANKL/RANK/TRAIL system acts on specific cell surface receptors, which are then able to transmit their signals to other intracellular components and modify gene expression. Cytokine production and activation of their receptors induce mechanisms to recruit monocytes and neutrophils as well as endothelial cells. Data support the role of an increased OPG/RANKL ratio as a possible marker of progression of endothelial dysfunction in metabolic disorders in relationship with inflammatory marker levels. We review the role of the OPG/RANKL/RANK triad in vascular function as well as molecular mechanisms related to the etiology of vascular diseases. The potential therapeutic strategies may be very promising in the future.

## 1. Introduction

Among the numerous molecules being studied for their potential utility as biomarkers of cardiovascular diseases (CVD), much attention is being given to the superfamily of tumor necrosis factor (TNF) receptors. Members of this family include osteoprotegerin (OPG) and its ligands, which are receptor activators of nuclear factor kappa-B ligand (RANKL) and TNF-related apoptosis-inducing ligand (TRAIL). TRAIL is a member of the TNF superfamily (TNFSF) and interacts with members of the TNF receptor superfamily (TNFRSF) [1,2]. OPG expression is regulated both positively and negatively by a wide array of factors, such as TNF and glucocorticoids. TNF is a central pro-inflammatory cytokine that controls the expression of numerous signaling pathways implicated in the progression of immunological reactions in relationship with the development of various diseases—vascular and metabolic diseases. Increased OPG production represents an early event in the development of diabetes mellitus and possibly contributes to diseases associated with endothelial cell (EC) dysfunction. The plasma OPG level is significantly coupled with endothelial function and the OPG serum level has a significant and independent predictive value for metabolic syndrome as a standard for cardiovascular risk in osteoporotic patients [3].

The balance between bone breakdown and reformation is modulated to a large extent by the secreted soluble receptor OPG. Recent studies have elucidated the crosstalk between ECs and osteoblasts during osteogenesis, thus connecting angiogenesis with osteogenesis. A relationship between bone regulatory proteins and vascular biology is now proposed. It has been demonstrated that OPG may mediate vascular calcification. Vascular calcification is a risk factor of cardiovascular and all-cause mortality in diseased patients. However, the cellular mechanisms involved in the links between vascular calcification and cardiovascular disease are mainly unknown, but growing evidence suggests that the RANK/RANKL/OPG triad may play a significant role in vascular calcification. In this article, we review the role of the OPG/RANKL/RANK/TSP/TRAIL system in endothelial metabolism and function as well as molecular mechanisms involving OPG related to the development of disease. New investigations are crucial to improving our knowledge in this area.

## 2. The OPG/RANKL/RANK/TRAIL System: Structures, Localization, and Characterization

OPG is a cytokine of the TNF receptor superfamily. It was named OPG because of its protective effects in bone (in Latin, “os” is bone and “protegere” is to protect). OPG is also known as osteoclastogenesis inhibitory factor (OCIF) or TNF receptor superfamily member 11b: (TNFRS11B). OPG is encoded by the TNFRSF11B gene. RANKL (TNFSF11) and RANK (TNFRSF11A), a receptor ligand pair of the TNF receptor superfamily, have emerged as the key molecular pathway in bone metabolism. (Figure 1).

Biochemically, OPG is a basic secretory glycoprotein composed of 401 amino acids (aa) with a monomeric weight of 60 kiloDaltons (kD). It is then assembled at the cys-400 residue in the heparin binding domain to form a 120 kD disulfide-linked dimer for secretion. OPG contains seven structural domains, which influence its biological activities in specific ways. Prior to secretion of the monomeric and dimeric forms of OPG, the 21 aa signal peptide is cleaved from the N-terminal, rendering a 380 aa mature OPG protein. Subsequently, circulating OPG exists either as a free monomer of 60 kD and a disulfide bond-linked homodimer form of 120 kD or as OPG bound to its ligands, RANKL, and TRAIL.

RANKL is a transmembrane protein, but a soluble form (soluble RANKL is sRANKL) also circulates in the blood. RANKL binds as a homotrimer to RANK on target cells, which triggers activation of nuclear factor κB (NF-κB). A key preliminary step in downstream signaling after RANKL ligation to RANK is the binding of TNF receptor-associated factors (TRAFs: 2,5,6) to specific sites in the cytoplasmic domain of RANK. TRAFs 2, 5, and 6 all bind to RANK. Several signaling pathways are activated by RANK/TRAF-mediated protein kinase signaling, such as NF-κB kinase (IKB)/NF-κB and activator protein-1, AP-1. Recently, it has become increasingly clear that these signaling pathways are present in various cells during vascular calcification [4].

OPG binds RANKL through its N-terminal cysteine-rich domains (CRD). The extracellular region of OPG consists of four CRDs, and each domain contains topologically distinct modules. CRDs are sufficient to inhibit RANKL [5]. Human RANK consists of 616 aa. These aa are divided into a C-terminal cytoplasmic domain of 383 aa, an N-terminal extracellular domain of 184 aa, a signal peptide of 28 aa, and a transmembrane domain of 21 aa, which contains four cysteine and two N-glycosylation sites. RANKL generates multiple intracellular signals by binding to RANK-TRAIL. TRAIL and its associated receptors exhibit broad tissue distribution. TRAIL mRNA and protein have been found in vascular smooth muscle cells (VSMCs) and ECs. TRAIL is expressed as a type II transmembrane protein. TRAIL also exists physiologically in a biologically active soluble homotrimeric form. TRAIL, also known as Apo2 ligand, is detectable in the serum under physiological conditions. TRAIL in its soluble form is detected at concentrations of 10–100 pg/mL in the serum/plasma. TRAIL can bind up to five distinct receptors to activate complex signaling pathways. OPG has also been noted to bind to TRAIL. An essential role of the TRAIL/TRAIL-R system is in the regulation and modulation of apoptosis. TRAIL may have a dual function in the immune system by being able to kill infected cells and by participating in the pathogenesis of multiple infections [6]. Interestingly, it has been suggested that TRAIL may also play a role in atherosclerotic plaque development. TRAIL is expressed in atherosclerotic lesions with increased levels seen at vulnerable plaque sites. Recent results suggest that the elevated levels of TRAIL present in atherosclerotic plaque could be harmful by intensifying the inflammatory response and reinforcing plaque formation. Some laboratories demonstrated increased apoptosis in TRAIL-treated EC, while other groups have shown increased survival and proliferation of these cells in response to TRAIL. It appears that TRAIL has pleiotropic effects within the vasculature [7]. In the literature, a number of studies have shown that soluble TRAIL is able to induce the activation of some signal transduction pathways, promoting the survival of VSMCs and ECs. TRAIL also exerts a protective effect on the endothelium through its anti-inflammatory properties and the production of local nitric oxide (NO) [8]. Recently, there has been outstanding progress in the development of novel formulations to increase the circulatory half-life of TRAIL, thus improving the biological attributes of TRAIL-based therapies.

Greater interest has been shown in improving our understanding of the interaction mechanism between RANKL and OPG, as manipulation of the OPG/RANKL ratio could be the basis for the development of new therapeutics. Besides binding RANK, OPG is able to interact with TRAIL and induces apoptosis of tumor cells through the cell-surface receptors death receptor 4 (DR4) and DR5 in the TNFRSF family [9]. The binding mode to RANKL was determined from computational docking and molecular dynamics simulations [10,11]. OPG is expressed in vivo by ECs, VSMCs, and osteoblasts. OPG has been detected by immunohistochemistry in aortic and coronary atherosclerotic plaques within or in the proximity of VSMCs [12,13]. 

OPG is released under basal conditions by ECs upon stimulation with inflammatory cytokines, hormones, and various circulating compounds. TNF-α and interleukin (IL)-1β were found to increase OPG levels. Within ECs, OPG is associated with von Willebrand factor (vWF) within secretory granules called Weibel-Palade bodies (WPBs). The size of vWF multimers can be controlled by the glycoprotein thrombospondin-1 (TSP-1). TSP-1 acts from within the endoplasmic reticulum to activate nuclear factor-E2-related factor 2 (Nrf2), inducing a protective antioxidant defense response against lipotoxic stress [14].

In vitro experiments show that OPG can be produced and also released by blood cells such as neutrophils and stem cells. Neutrophils produce OPG and IL-17. IL-17 increases the recruitment of neutrophils at the site of inflammation and influences the production of various proinflammatory mediators [15,16]. A recent study reported a significant elevation of circulating OPG in septic patients with different levels of severity and in those who progressed to acute kidney injury; OPG thus appears to be a reliable biomarker [17].

OPG is also released by various kinds of stem cells. Vascular stem/progenitor cells (VSCs) are an important source of all types of vascular cells needed to build and repair blood vessels. There are several types of VSCs, including endothelial progenitor cells (EPCs), smooth muscle progenitor cells (SMPCs), mesenchymal stem cells, and adipose-stromal cells (ASCs). ASCs are one of the most important and promising cell sources in the field of regenerative medicine. Recently, the production of OPG by ASCs and its role in vascular pathophysiology were examined. It was demonstrated that OPG generated apoptosis of EPCs by inducing oxidative stress. This effect was mediated by syndecan-4 and oxidative stress. Syndecans are plasma membrane proteoglycans, and oxidative stress alters syndecan-distribution in tissues. OPG-induced apoptosis was abolished by reactive oxygen species (ROS) scavengers such as *N*-acetylcysteine and the NADPH oxidase (NOX) inhibitor, diphenyleneiodonium. OPG increased ROS production through activation of NOX-2 and NOX-4 and triggered phosphorylation of ERK-1/2 and p38 MAPK [18]. In ASCs, the link between oxidative stress, apoptosis, and OPG was recently confirmed. Hydrogen peroxide (H_2_O_2_) significantly increased OPG production by ASCs in vitro. OPG production by ASCs transplanted into ischemia–reperfusion-injured hearts was also observed. It was suggested that OPG is one of the protector factors released by ASCs contributing to ASC-mediated cardioprotection. However, the mechanisms of OPG-mediated cellular protection have not yet been completely elucidated [19]. 

A number of polymorphisms in the promoter region of the OPG gene have been described in different diseases. Each polymorphism has been evaluated in specific diseases. Various studies have been designed to evaluate the association between polymorphisms of the OPG gene, the serum OPG level, and the advance of atherosclerosis associated (or not) with rheumatoid arthritis (RA). One polymorphism of the TNFRSF11B gene has been coupled with the presence of coronary atherosclerosis in patients with RA [20].

Finally, elevated OPG levels are associated with markers of inflammation, endothelial dysfunction, oxidative stress, and CVD [2].

## 3. Interactions between OPG/RANKL/RANK and Endogenous Factors in the Heart: Incidences on Metabolism and Functions of Endothelial Cells.

The human heart consists of a variety of cell types with fibroblasts and other connective tissue cells being the most abundant [21]. The remaining cell mass consists of cardiomyocytes, EC, VSMCs, mast cells, and immune-related cells. However, CM mass is approximately 25 times that of EC mass. Cardiomyocytes are the major consumers of oxygen in the heart and account for approximately 75% of normal myocardial volume, and there is at least one capillary adjacent to every cardiomyocyte. Cardiomyocytes are outnumbered ≈3:1 by ECs in the microvasculature and small vessels in the myocardium [22]. 

The endothelium is one of the largest “organs” in the body and probably also one of the most heterogeneous. The endothelium includes a large collection of EC subtypes differing in phenotype, function, and location. The different ECs adapt the flux through the metabolic pathways in relationship with the specific energy sources, the redox balance, and precise metabolisms [23]. 

In healthy adults, ECs are quiescent and exert a barrier function and maintain tissue homeostasis. They have the capacity to form new vasculature in response to angiogenic factors induced by injury and/or pathological conditions, such as hypoxia or tissue damage. In the myocardium, capillary EC, in situ, are able to change shape against a continuous flow and adapt to the contractive environment [23]. Metabolic activities in ECs are different from those in other cells, whose cellular bioenergetics are linked to oxidative mitochondrial metabolism. ECs can alter their phenotypes and switch among different states—migrating, proliferative, and quiescent.

ECs of the microcirculation are fundamental for myocardial function, which largely depends on the ratio between energy metabolites received from the coronary circulation and their use by cardiomyocytes. Endothelial tissue originating from different organs may differ in terms of its metabolic profile. ECs have a smaller number of mitochondria than other cell types and therefore consume lower amounts of oxygen. Likewise, the intracellular distribution of mitochondria varies among the different EC and suggests their important regulatory roles in cellular homeostasis. ECs generate up to 85% of their ATP through aerobic glycolysis. Interestingly, the rate of glycolysis differs in EC subtypes. Arterial ECs are more oxidative, whereas microvascular ECs are more glycolytic [24]. Despite the adaptation of ECs to use glucose, they also need other metabolic sources of energy to carry out their functions. Fatty acids (FAs) catabolized by fatty acid-beta-oxidation (FAO) are an important fuel for ECs during sprouting [25]. The regulation of FAO is modulated by a variety of influences, including the peroxisome proliferator-activated receptor (PPARs) family of transcription factors. High FA levels activate PPAR-α and thereby increase FAO.

The heart is capable of remodeling metabolic pathways in chronic pathophysiological conditions, which results in modulations of myocardial energetics and contractile function. Because high-energy phosphate storage within the cardiomyocyte is minimal and only sufficient to maintain the heart beat for a few seconds, a strong coupling of ATP production and heart contraction is necessary for normal cardiac function [26]. To preserve its function, the heart, a high-energy organ, exhibits “plasticity” in its ability to use multiple substrates for energy production, including FAs, carbohydrates, and ketone bodies. In cardiomyocytes, FAs are predominantly used as an energy source. 

In the normal heart, nearly 70% of ATP is produced from FA oxidation. The heart has a high demand for FA, but it has a restricted capacity to synthesize FA and thus depends on an exogenous source of FA. FAs are delivered inside the capillary lumen through the hydrolysis of triglyceride-rich lipoproteins by lipoprotein lipase. In this context, ECs play a key function. In the heart, ECs express the FA-binding proteins FABP4 and FABP5, which transport FAs across the endothelium [27]. Vascular endothelial growth factors-B (VEGF)-B secreted by cardiac and skeletal muscle and brown adipose tissue produces the FA transport proteins via VEGF receptor 1 in capillary ECs [28].

Endothelial senescence could play a significant role in cardiac diseases such as hypertrophy, and in this state, it is well established that cardiac metabolism undergoes reprogramming. These changes are characterized by increased glucose metabolism and decreased FAO. Concerning the impact on glucose metabolism, the upregulation of glucose uptake associated with decreases in overall ATP synthesis by oxidative metabolism is observed, and glycolysis is thus increased [29]. While increased glucose utilization appears to be beneficial for the failing heart, decreased FA supply to the hypertrophied and failing heart seems to be detrimental. The shift in substrate preference to glucose in pathological hypertrophy was considered adaptive given the theoretically higher oxygen efficiency of ATP synthesis from glucose [30]. In conclusion, there is crosstalk between the endothelium and cardiomyocytes, and metabolic maladaptation can impair cardiac function.

An interesting link exists between ATP/adenosine metabolism and the functions of the OPG/RANK/RANKL triad. Adenosine may either be released from the intracellular space by exocytosis or may generate by the enzymatic breakdown of extracellular ATP. Adenosine exerts a variety of physiological effects by binding to cell surface G-protein-coupled receptor subtypes A1, A2a, A2b, and A3. In various organs, the role of adenosine is to prevent tissue injury; it acts as a cytoprotective modulator. In vitro, in a human osteoprogenitor cell line, it has been shown that adenosine and adenosine receptor agonists inhibited OPG secretion [31]. In rheumatoid arthritis (RA) patients, the OPG/RANKL ratio is elevated in blood samples and the A3AR is over-expressed in inflammatory cells. These data reflect in these patients the autoimmune inflammatory disease [32,33]. RA accelerates atherosclerosis and increases occurrence of vascular diseases. The development of metabolomic analysis is able to clarify the interactions between inflammation and metabolic changes underlying many diseases, such as RA. 

ECs produce high levels of OPG in response to stimulation by lipopolysaccharides or other activators [34,35]. Yet, OPG affects the cytoskeletal organization of ECs via its molecular effects. In vitro, treatment of ECs with OPG induced the reorganization of the cytoskeleton of endothelial colony-forming cells (ECFCs). ECFCs, also termed late-outgrowth ECs, are a well-defined circulating EPC type with an established role in vascular repair. OPG induced activation of αVβ3 integrin and the regulation of its ligand, protein-disulfide-isomerase. In addition to its role in cell migration, αVβ3 integrin promotes the survival of stimulated ECs [36]. In this context, heparan sulfate proteoglycans (HSPGs) may regulate OPG bioavailability. Proteoglycans of the syndecan family are involved in modulating integrin-mediated tight adhesion of leukocytes to the endothelium. On the other hand, HSPGs immobilize chemokines on luminal ECs, thus protecting them against mechanical or hemodynamic variations [37]. 

Abnormalities of HSPGs have been found in mitral valve degeneration. Isolated human valve ECs exhibited evidence of endothelial to mesenchymal transition (EndMT) [38]. Data reported in a recent study validated the hypothesis that OPG might represent a novel actor in the progression of this disease. The overexpression of OPG has been demonstrated during EndMT and linked to autocrine effects characterized by the increased production of ROS. OPG interferes with correct valve endothelial function by increasing proteoglycan and matrix metalloproteases (MMPs) levels [39].

Factors like RANKL, RANK, and OPG are involved in the process of atherosclerosis by altering lipid metabolism. High Density Lipoproteins (HDL) subclasses may be indirect players in the process of the atherosclerotic plaque through the regulation of the expression of genes that encode pro- and anti-calcifying proteins. Data suggest that HDLs protect against the progression of atheroma through mechanisms involving the regulation of genes. In this context, the role of the superfamily of TNF receptors is suggested, and a member of this family—such as OPG—is suggested. In vitro, the incubation of myofibroblasts with HDL for 24 and 48 h resulted in a time-dependent increase in OPG secretion [40]. 

Concerning the glucose metabolism, the uptake of extracellular glucose is regulated by the transmembrane glucose gradient and the activity of glucose transporters in the plasma membrane. Insulin leads to the relocation of glucose transporters to the plasma membrane with a subsequent increase in capacity for glucose transport. Once in the cell, free glucose is rapidly phosphorylated by hexokinase to form glucose-6-phosphate (G6P). G6P is used for glycogen synthesis or may undergo glycolysis to pyruvate. As we reported, FAs are the preferred substrate for the myocardium; however, during ischemia, glucose becomes the primary source of energy for the myocardium. Its metabolism avoids the toxic end-products such as oxygen free radicals (OFR). Patients with diabetes mellitus have impaired uptake of glucose. In diabetic situations, ECs cannot switch the excess glucose, and glycolytic intermediates drift to side pathways, overall increasing oxidative stress. OFR, such as superoxide, react with nitric oxide (NO) to yield peroxynitrite [41,42].

The growth factor system exerts various effects (on glucose metabolism in particular) in cells of the vasculature through both endocrine and autocrine/paracrine mechanisms. The growth factor system, which includes VEGFs and platelet-derived growth factor (PDGF), a basic fibroblast growth factor, is a key regulator of EC permeability and metabolism. VEGFs and PDGF influence human EC metabolism via Ca^2+^ signaling mechanisms. Mechanisms underlying the endothelial actions of these factors are multiple, and they have been shown to participate in the initiation and development of atherosclerosis. In vascular cells, PDGF upregulates OPG expression [43].

Statins have an impact on endothelial function by preventing oxidized LDL-induced reduction of NO production and increased NO synthesis. Statins also diminish chronic inflammation by reducing PDGF responsiveness and inhibiting not only smooth muscle cell proliferation but also monocyte chemotaxis and migration [44]. 

## 4. OPG/RANKL/RANK and Vascular Signaling

The location of vascular ECs links them to various types of mechanical forces—hydrostatic pressure, wall tension, and shear stress. Shear stress regulates cellular functions and gene expression, thus showing the involvement of potential sensors and effectors. Intermediate responses to shear include transcriptional activation of NF-κB target genes [45,46]. To our knowledge, nothing is known about the specific effect of shear stress on the expression of OPG in vascular endothelium. In contrast, it has been reported that shear stress upregulated OPG expression in osteocytes, downregulated the effect of IL-17A on RANKL and TNF-α expression, and attenuated IL-17A-activated osteoclastic differentiation [47].

With advancing age, the phenotype of VSMC and EC changes. Various stimuli promote the development of advanced atherosclerotic lesions. The renin-angiotensin system (RAS) plays a central role in the pathogenesis of vascular alterations and atherosclerosis in the elderly. RAS and its primary mediator, angiotensin-II (Ang II), have a direct influence on the progression of the atherosclerotic process via effects on endothelial function and inflammatory processes. Therapies that block Ang II receptor type 1 induce vascular protection and thus reduce the incidence of cardiovascular events [48]. The stimulation of Ang II has been reported to increase the expression of VEGFs through the activation of Ang II receptor type 1. Members of VEGF family, VEGF-A and VEGF-B, are involved in vascular inflammation and remodeling through increased proinflammatory and angiogenic mechanisms. It was demonstrated that OPG enhanced the proangiogenic effect of VEGFs. Additionally, OPG protects EC from apoptosis induced by growth factor withdrawal [49].

In a recent study, atheroma samples obtained from patients undergoing carotid endarterectomy were cultured with and without an Ang II type 1 receptor (ATR1) antagonist, irbesartan. Irbesartan reduced concentrations of cytokines, IL-6, IL-8, and OPG in both atheroma and primary vascular cell culture supernatants. In these experimental conditions, which used human dermal microvascular ECs, ATR1 blockade with irbesartan also led to a decrease in the expression of extracellular signal regulated kinases, ERK1 and ERK2. Similarly, a more recent study in mice showed that RANKL-induced ERK1/2 phosphorylation was suppressed by another ATR1 inhibitor, Losartan, suggesting a convergence of RANKL and angiotensin signaling at the level of ERK1/2 regulation [50,51]. OPG activates ERK 1/2, which has been linked to angiogenesis. 

## 5. OPG/RANKL/RANK and Regulation of Angiogenesis

It is now accepted that RANK and its ligand RANKL are involved in endothelial physiology. The RANKL/RANK system plays an active role in pathological angiogenesis and inflammation in addition to its role in cell survival. Growth factors can act on specific cell surface receptors that are then able to transmit their growth signals to other intracellular components and modify gene expression. One example of a protein growth factor with specific properties on EC is VEGF. VEGF up-regulates the expression of RANK and increases angiogenic responses of ECs to RANKL. Moreover, blocking PI3-kinase reversed the RANKL-induced survival effect on ECs [52]. RANK, in response to the paracrine stimulus of RANKL, may play an important role in maintaining EC integrity through the PI3-kinase/Akt signal transduction pathway. In the endothelium, PI3-kinase/Akt signaling is triggered by VEGF and hormones such as insulin [53]. Findings suggest that OPG regulates at least two distinct pathways—one that induces cell proliferation via ERK signaling and another that induces angiogenesis via Src signaling [54]. Bone is a highly vascularized tissue reliant on the close spatial and temporal connection between blood vessels and bone cells to maintain skeletal integrity. An intricate connection between osteogenesis and angiogenesis exists. Decreasing activity of osteoblasts leads to osteoporosis, and crosstalk between osteogenesis and angiogenesis has been shown to play a vital role in bone regeneration [55,56]. Accumulating evidence supports the role of exosomes secreted EPCs in stimulating angiogenesis, which is closely coupled with osteogenesis [57]. Taken together, these results suggest that RANK is important for the maintenance of endothelial integrity in association with metabolic adaptations.

## 6. OPG/RANKL/RANK and Inflammation

Numerous studies support the role of OPG in promoting inflammation. In the pro-atherosclerotic apolipoprotein knock-out mouse, it was demonstrated that a deficiency of OPG was associated with increased development of atherosclerosis [58]. In vitro studies confirmed that OPG plays an important role in inflammatory cell chemotaxis. As previously stated, OPG stimulates changes in vascular smooth muscle cells and endothelium, which are usually reported in atherosclerosis, by promoting apoptosis and matrix metalloproteinase release. RANKL significantly increases the activity of MMPs in VSMCs. OPG neutralizes the effect of RANKL on the induction of MMP activity in VSMCs by inhibiting its binding to RANK [59,60]. One of the key steps during inflammation is leukocyte infiltration, which, for neutrophils and monocytes, is controlled chiefly by chemokines. The production of these chemokines is regulated by iNOS-derived NO [61]. OPG has been proposed as a marker of endothelial dysfunction in relationship with the inflammatory process. OPG induces the expression of intercellular adhesion molecules, such as vascular adhesion molecule-1 (VCAM-1) and E-selectin, on ECs and thereby promotes leukocyte adhesion, an early step in EC dysfunction, thus supporting the pro-atherosclerotic role of OPG. These local actions, which influence the velocity of leukocyte recruitment from the blood to the tissue, contribute to the multifunctional role of various modulators, such as HSPGs in inflammation [62]. The release of OPG is significantly triggered by the culture of ECs with inflammatory cytokines and leads to the expression of EC adhesion molecules, thereby contributing to the transmigration of monocytes and lymphocytes into the intima of the vessel wall [63]. Cytokine production and activation of their receptors induce mechanisms to recruit monocytes and neutrophils. Therefore, blocking pro-inflammatory interleukins is considered a prime target in the management of some diseases. New molecules represent potential therapeutic strategies. Canakinumab and evolocumab, human monoclonal antibodies that target interleukin-1β, have anti-inflammatory effects and have been approved for clinical use in various disorders [64]. Sarilumab and tocilizumab are human monoclonal antibodies against IL-6 receptor-α (IL-6R α) [65]. Activation of IL-6R is protective and regenerative in some types of cells, but IL-6 signaling via the soluble IL-6R is rather pro-inflammatory. Interestingly, it was recently reported that in human breast cancer cell lines, IL-1β induced OPG secretion, indicating a novel role for OPG as a mediator of inflammation-promoted breast cancer progression. The increased cellular invasion promoted by IL-1β and OPG involves MMP3 induction [66]. (Figure 2).

Numerous studies have demonstrated that endothelial and inflammatory cells express RANKL. RANKL significantly increases the activity of MMP in VSMCs, and OPG neutralizes the effect of RANKL on the induction of MMP activity in VSMCs by inhibiting its binding to RANK [60]. Interestingly, in ECs, a relationship has been demonstrated between oxidative stress and RANKL. Incubation of ECs with oxidized low density lipoprotein (OxLDL) and other pro-oxidant molecules, such as H_2_O_2_, increased RANKL in a dose-dependent manner. Thus, oxidative stress-regulated RANKL expression appears to be a general phenomenon [67]. The observation that OxLDL stimulated RANKL expression in different vascular cell types revealed one of the processes whereby vascular alterations occurred in patients with an elevated OxLDL level [68]. RANKL was recently demonstrated to potently activate human neutrophil degranulation via the binding to its transmembrane receptor RANK, and RANKL was also shown to be protective against post-ischemic inflammation. Anti-RANKL IgG was shown to exert a potential direct effect on the activation of cardioprotective RISK and SAFE intracellular pathways [69,70]. In the presence of fibroblast growth factors (FGFs), such as FGF21, the expression levels of proteins, including RANKL, were down-regulated, whereas the expression of OPG increased. FGF21 was reported to play a protective role against oxidative stress-related endothelial damage, atherosclerotic plaque formation, and ischemic injury of cardiomyocytes [71,72]. Adaptive immunity appears crucial for endothelial functions. There is growing evidence that innate and adaptive immunity are critical for the properties of the endothelium. In this field, growth differentiation factor 11 (GDF11), a secreted member of the transforming growth factor beta (TGF-β) superfamily, contributes to the regulation of angiogenesis [73,74,75].

Concerning adaptive immunity, it has been reported that following administration of GDF11, changes in cardiomyocytes are associated with activation of SMAD2, the ubiquitin-proteasome pathway [76]. Finally, it is difficult to overstate the importance of the RANKL–RANK–OPG system with respect to understanding how the TGF-superfamily is controlled.

## 7. OPG/RANKL/RANK and the Proteasome

Alterations in the ubiquitin-proteasome system (UPS) contribute to the pathogenesis of several diseases, including cancer, neurodegenerative and immune diseases, and atherosclerosis in association with processes of endothelial dysfunction. In vascular cells, a fundamental role has been assigned to the interaction between the UPS and the oxidative stress response. Several data concern the participation of the UPS in the regulation of eNOS expression and activity [77]. The UPS is also an important molecular mechanism involved in regulating vascular and EC aging [78]. Increased ubiquitin staining and reduced proteasome activities have been described in the pathogenesis of congestive heart failure. Several mechanisms are involved in the decline of proteasome activities in these pathological hearts [79]. Interestingly, in experimental models of heart failure, significantly increased mRNA expression of OPG was noted in both the ischemic and non-ischemic myocardium compared with that in subjects without heart failure, suggesting a potential role of OPG in the adaptation of the myocardium to the failure. The OPG/RANK/RANKL axis appears to be activated within the myocardium in the rat model of post-infarction heart failure, implying a potential role for the RANKL/RANK interaction in the pathogenesis of this cardiac disease [80,81]. Therefore, the proteasome pathway in relationship with the OPG/RANK/RANKL axis may represent an effective therapeutic target for the prevention and treatment of cardiac diseases.

## 8. OPG/RANKL/RANK and Cellular Senescence

Aging-related endothelial dysfunction involves increased oxidative stress, the activation of inflammatory pathways, and impaired regeneration of ECs. Multiple mechanisms responsible for cellular senescence have been proposed, among which the shortening of telomeres associated with the increased oxidative stress appears to be the most important [82]. It is now recognized that OPG participates in protection against atherosclerosis and vascular calcification. There is good evidence to suggest that OPG is involved in cell survival and proliferation [83]. Recent results demonstrate that irradiation-induced senescent tumor cells influence the tumor microenvironment by increasing the production of cytokines, such as OPG. OPG is also considered a survival factor for tumor cells by inhibiting tumor cell apoptosis [84]. OPG is able to induce the activation of the angiogenic signaling pathways in ECs. In addition, OPG has pro-inflammatory effects that could be mediated by the activation of the NF-κB pathway and expression of specific genes [85].

## 9. OPG/RANKL/RANK and Vascular Calcification

Arterial calcification results from a highly regulated process that shares many similarities with bone formation. The nature of the cells responsible for the formation of arterial calcification is not precisely known. The development of vascular calcification is an active and complex process linked with a multitude of signaling pathways [86]. SMC have been shown to have osteochondrogenic potential. However, recent evidence suggests that various vascular cells—and particularly the pericytes—play a role in this process. Resident vascular pericytes may have a protective effect against the development of vascular calcification. They participate in association with other cells such as monocytes/macrophages in regulating the balance of mineral formation [87].

Moreover, higher pericyte cell density was noted in asymptomatic lesions, suggesting that pericytes could be actively involved in plaque stability. It has been suggested that exposure to inflammatory atherosclerotic stress induces pericytes. Pericytes could be involved in the onset of the mineralized structure in plaques and in the secretion of OPG. Human pericytes secrete elevated amounts of OPG in comparison to SMCs and ECs [88,89]. One of the key functions of pericytes in both skeletal and cardiac muscle is in the modulation of angiogenesis through the promotion of EC survival and migration. Recent evidence suggests that in response to injury, pericytes are also able to modulate local tissue immune responses via several independent pathways. In this area, the OPG/RANK/RANKL axis in association with the functions of pericytes may be involved in vasculogenesis. OPG-mediated angiogenesis involves the MAPK and Akt signaling pathways [90,91]. The ability of pericytes to enhance myocardial repair has been demonstrated. However, the underlying mechanisms are less clear than those in skeletal muscle [92]. Injured hearts into which pericytes were transplanted exhibited significant attenuation of the post-injury decline in cardiac pump function. These effects are associated with decreased inflammation and increased angiogenesis [93]. OPG appears to afford protection against vascular calcification since OPG−/− mice developed spontaneous arterial calcification, and depleting OPG in ApoE-/- mice increased atherosclerotic lesion progression and calcification [94]. Concerning the incidence of RANK/RANKL on vascular calcification, these factors have roles in both promoting and inhibiting this process. There are many factors impacting vascular calcification, which is a complex process in relation to an early stage of chronic kidney disease (CKD). It is recognized that RANKL increases vascular smooth muscle cell calcification by binding to RANK and increasing BMP4 production through activation of the alternative NF-κB pathway [95]. As RANKL is thought to promote vascular muscle cell calcification, RANKL inhibition by specific agents, such as denosumab, have been tested for their ability to prevent vascular calcification [96].

As we previously reported, RANKL, RANK, and OPG are involved in the process of atherosclerosis by altering lipid metabolism. HDLs protect against the progression of atheroma through mechanisms involving the regulation of production of pro and anti-calcifying proteins. A correlation between the calcium score and phospholipids of HDL subclasses was described. Coronary artery calcification scores and lipid profiles are independent aspects of atherosclerosis, and only lipids may be biomarkers of coronary calcification during the asymptomatic stages of the disease [97]. 

Vascular calcification is an active cell-mediated process, and like osteogenesis, it involves the expression of bone-related proteins, such as alkaline phosphatase (ALP) and Runt-related transcription factor-2 (Runx2), which are initiators of bone mineralization and SMC differentiation. Calcified atherosclerotic lesions have been shown to express ALP [98]. Otherwise, high ALP levels are associated with an increased risk of cardiovascular events and mortality in patients. This link between ALP and the coronary artery calcium score is subject to several confounding factors, such as Glomerular Filtration Rate (GFR), an inflammatory status with mediators acting on endothelial function. It is now well-recognized that OPG is significantly associated with endothelial function and predicts early carotid atherosclerosis in patients with coronary artery disease (CAD). The carotid intima-media thickness (CIMT) is correlated with carotid atherosclerosis and is a significant predictor of cardiovascular events. The OPG levels are associated with the CIMT in CAD patients [99,100]. OPG has been proposed as a marker of endothelial dysfunction of early pathophysiological events. Among the tested 12 inflammatory markers, only the OPG had significant prognostic value in predicting the occurrence of atrial fibrillation (AF). The levels of OPG were significantly associated with incident AF [101,102].

In conclusion, it is evident that impaired bone metabolism has an important role in the development of vascular calcification. Bones and vessels have mutual changes in mineralization; this situation is called “the bone-vascular axis” and is associated with coronary diseases [103].

## 10. Summary

The existence of crosstalk between ECs and osteoblasts during osteogenesis was demonstrated, thus establishing a link between angiogenesis and osteogenesis. A relationship concerning bone regulatory proteins and vascular biology was proposed, and it was established that OPG mediates vascular calcification. The RANKL/RANK system plays an effective role in cell survival under normal and pathological conditions, angiogenesis and inflammation. A great effort was made to determine the mechanism of interaction between RANKL and OPG and the initiation of diseases. The manipulation of the OPG/RANKL ratio is the basis for developing new therapeutics.

New data showed the potency of innovative peptides as inhibitors of RANKL, and results provided useful information for the development of various therapeutics. Previously, we reported that RANKL inhibition by specific agents such as denosumab were tested for their ability to prevent vascular calcification [96]. Evidence suggests that inhibition of RANKL does not only induce an increase in bone mass and vascular calcification but also has anti-tumor effects [104]. RANKL and RANK are expressed on cells of the immune system—in particular, B cells and activated T lymphocytes. The expression of RANKL in cells of the immune system contributes to the pathogenesis of several autoimmune diseases, such as rheumatoid arthritis. In vitro, a Jak1/2 inhibitor, baricitinib, inhibits osteoclastogenesis by suppressing RANKL expression in osteoblasts [105]. While the role of the RANKL/RANK/OPG axis in bone remodeling has been greatly studied, the role of this triad in the central nervous system has only begun to arise. RANKL mRNA and RANK/RANKL expression are localized to the brain. Thus, the OPG/RANKL/RANK axis appears to play a role in controlling the central febrile response and inflammation in ischemic brain [69].

Concerning the potential clinical properties of TRAIL, the context is contradictory. Unlike serum levels of OPG, those of TRAIL are significantly lower in patients affected by or predisposed to CVD. Potentially, TRAIL is a “janus” molecule with two faces, the first able to induce apoptosis and stimulate inflammation and the second likely to promote cell survival and inhibit inflammation. These opposing effects depend on its concentration. The specific localization of the TRAIL receptor complex may be another mechanism involved in the TRAIL-induced anti-apoptotic signaling events. It was suggested that it would be useful to develop novel formulations to increase the circulatory half-life of TRAIL with the aim to improve the beneficial actions attributed to TRAIL in various therapies. Another future clinical direction concerns the genomic analysis of certain proteins related to the inflammatory process and OPG signaling. For example, Ecto-5’-nucleotidase/CD73/NT5E, the product of the NT5E gene, is the dominant enzyme in the generation of adenosine from the degradation of ATP. As we previously reported, in a human osteoprogenitor cell line in vitro, it has been shown that adenosine and adenosine receptor agonists inhibited OPG secretion [31]. CD73 is found in a variety of tissues including endothelium. The endothelial CD73 axis regulates hemostasis by converting the local environment from a prothrombotic ATP/ADP-rich state to an antithrombotic, adenosine-rich environment. Mutations in NT5E, which codes for Ecto-5’-nucleotidase (CD73), result in calcifications of the lower-extremity arteries in patients with a syndrome called CALJA (calcification of joints and arteries) [106]. Recent studies suggest that active processes contributing to vascular calcification are compensated by calcification inhibitors. Genetic or pharmacological interventions interfering with CD73 activity may prove useful in various diseases [107]. CD73 inhibitors such as adenosine 5’-α,β-methylene-diphosphate present promising potential as a therapeutic target [108]. Pharmacogenomics is an area where genomic discoveries are able to improve clinical care.

## Figures and Tables

**Figure 1 ijms-20-00705-f001:**
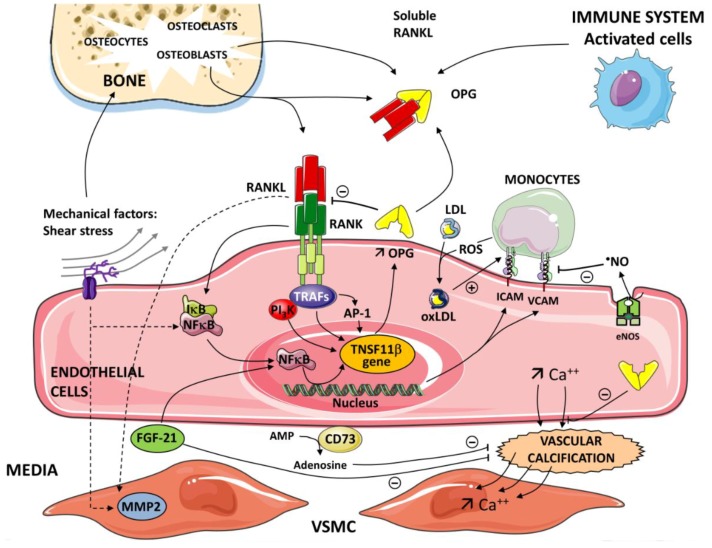
Critical role of the nuclear factor kappa-B/nuclear factor kappa-B ligand/osteoprotegerin (RANK/RANKL/OPG) axis in the pathogenesis of inflammatory processes and vascular calcification. OPG is produced by different cells—activated cells (immune system), osteoblasts in bone. The inflammatory cells and immune cells up-regulate expression of receptor activator of the RANKL. A soluble form of RANKL, sRANKL, also circulates in the blood. The interaction between RANK and RANKL initiates a signaling and gene expression cascade, activating the transcription factor NF-κB. OPG binds to RANKL and prevents the RANKL/RANK interaction. Tumor necrosis factor (TNF) receptor-associated factors (TRAFs 2,5,6) to specific sites are present in the cytoplasmic domain of RANK. Subendothelial retention of low-density lipoprotein (LDL) and its oxidative modification (OxLDL) represent the initial event in atherogenesis. Reactive oxygen species (ROS) generated by monocytes contribute to the level of oxidation of LDL. OxLDLs induce endothelial cell (EC) expression of adhesion molecules intercellular adhesion molecule-1 (ICAM-1) and vascular adhesion molecule-1 (VCAM-1). Nitric oxide (NO) generated in the endothelium by the catalytic action of the enzyme nitric oxide synthase (eNOS) reduces the endothelial expression of ICAM-1 and VCAM-1. In the nucleus of ECs, via NF-κB and AP -1, OPG induces the expression of ICAM-1 and VCAM-1 and promotes leukocyte adhesion, an early step in ECs dysfunction. Various pathways and mediators are involved in vascular calcification depending on the etiology of the atherosclerosis. Vascular calcification is an active cell-regulated process of mineralization implicating matrix mineral metabolism. Sensors and effectors associated with shear stress regulate cellular functions and gene expression via the activation of NF-κB target genes. Osteogenic differentiation of vascular smooth muscle cells (VSMC) plays a pivotal role in the progression of vascular calcification. RANK-RANKL-OPG and other regulatory proteins are major pathways in the progression of vascular calcification. Fibroblast growth factor21 (FGF21) and Ecto-5’-nucleotidase (CD73) contribute to the regulation of this calcification. FGF21 protects the vascular system by limiting VSMC calcification. CD73 hydrolyses extracellular AMP to adenosine. Adenosine has been shown to play a protective role against calcification.

**Figure 2 ijms-20-00705-f002:**
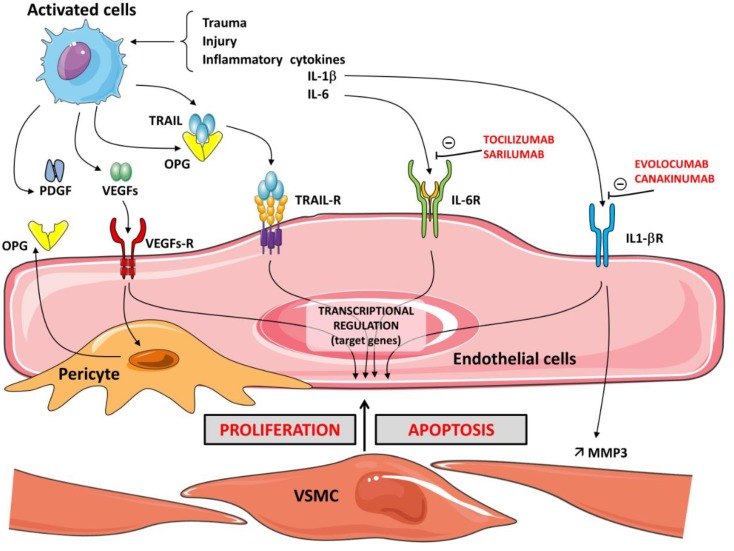
Schema illustrating the relationship between the OPG/TRAIL/TRAIL-R system, pericytes, growth factors, and the cytokines IL-1 and IL-6 on the balance between proliferation and apoptosis of vascular smooth muscle cells (VSMC). In the presence of inflammatory cytokines IL-1β or IL-6 and trauma or injury, activated cells express OPG. Activation of cytokine receptors IL-1R and IL-6R induces the recruitment of monocytes and neutrophils. The growth factor system, which includes vascular endothelial growth factors (VEGFs) and PDGF, influences the proliferation (angiogenesis) and OPG expression in vascular cells. Associated with the microvasculature, pericytes secrete elevated amounts of OPG. OPG also acts as a receptor for TRAIL. TRAIL binds to its receptors, TRAIL-Rs. The cellular actions of TRAIL are tightly regulated in a balance between the apoptosis and proliferation of vascular cells. TRAIL and RANKL increase matrix metalloproteinase (MMP) activity, leading to degradation of the extracellular matrix. The cellular actions promoted by IL-1β induce MMP-3 production. MMP-3 is constitutively expressed in EC and VSMC. The human monoclonal antibodies canakinumab and evolocumab, which target IL-1β, have anti-inflammatory effects. Sarilumab and tocilizumab are human monoclonal antibodies against the IL-6 receptor-α. Intracellular signaling of IL-6 in response to receptor activation is complex—anti-inflammatory protective but pro-inflammatory for the immune response.

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
