# Peer review of "The Role of Osteoprotegerin and Its Ligands in Vascular Function"

_ijms, 2019, doi:10.3390/ijms20030705_

Round 1

Reviewer 1 Report

The review exhaustively describes the role of osteoprotegerin and its ligands in vascular function also investigating the potential therapeutic implications. The topic is well addressed and comprehensive, sometimes a bit lengthy but never repetitive.

I recommend its acceptance with minor revisions:

the manuscript could be enhanced adding more recent references;

the text would benefit from a revision of the English style by a native speaker;

I recommend a careful proofreading to avoid small errors and inhomogeneity in the text.

Author Response

REPLY to the Review Report ( Reviewer 1)

The authors thank the Reviewer for the careful review of our manuscript and for the interesting suggestions he/she made.

Reviewer’comment:

1.       The manuscript could be enhanced adding more recent references;

As suggested, we added the following references:

TLR2 Promotes Vascular Smooth Muscle Cell Chondrogenic Differentiation and Consequent Calcification Via the Concerted Actions of Osteoprotegerin Suppression and IL-6-Mediated RANKL Induction. Lee GL, Yeh CC, Wu JY, Lin HC, Wang YF, Kuo YY, Hsieh YT, Hsu YJ, Kuo CC. Arterioscler Thromb Vasc Biol. 2019 Jan 10

The Interplay of SIRT1 and Wnt Signaling in Vascular Calcification. Bartoli-Leonard F, Wilkinson FL, Langford-Smith AWW, Alexander MY, Weston R. Front Cardiovasc Med. 2018 Dec 18;5:183.

CD73 as a potential opportunity for cancer immunotherapy. Ghalamfarsa G, Kazemi MH, Raoofi Mohseni S, Masjedi A, Hojjat-Farsangi M, Azizi G, Yousefi M, Jadidi-Niaragh F. Expert Opin Ther Targets. 2019 Feb;23(2):127-142

This new review has been redacted in order to make the findings in specific fields more clear.

2.        The text would benefit from a revision of the English style by a native speaker;

As suggested, the manuscript is corrected by a native English speaker. The authors wish to thank Philip Bastable for English assistance.

3.        I recommend a careful proofreading to avoid small errors and inhomogeneity in the text.

 Our manuscript has been modified in order to avoid some errors.

Reviewer 2 Report

This manuscript by Luc Rochette et al describes the current knowledge of OPG and related pathways in regulating vascular functions. Overall, although the topic is relevant, it needs some focus and elaboration on specific topics in this manuscript.

Major points,

1 the whole manuscript is not concise and kind of discrete. For example, section 2 discussed a lot about the biochemical properties of OPG and its ligands, which seems redundant for a review paper.  On page 4/5, the authors describe the metabolic properties of ECs, which are dispensable for the topic they are discussing in other parts. Especially, I don’t see any description of OPG pathways here.

2 Quite significant proportion of references are missed in this manuscript. For example, line 42, line 465 the authors discussed the connection of angiogenesis and osteogenesis without any reference.

3 more recent works in this field are expected here.

Minor point

1 line 275-278 has a different format with other parts of the manuscript.

Author Response

REPLY to the Review Report (Reviewer 2)

The authors thank the Reviewer for the careful review of our manuscript and for the interesting suggestions he/she made.

Reviewer’comment:

1.       The whole manuscript is not concise and kind of discrete. For example, section 2 discussed a lot about the biochemical properties of OPG and its ligands, which seems redundant for a review paper.  On page 4/5, the authors describe the metabolic properties of ECs, which are dispensable for the topic they are discussing in other parts. Especially, I don’t see any description of OPG pathways here.

We deleted a part of basic information which is not directly related to the aim of this review:

P5: [22]. The endothelium has a large surface area (approximately 350 m2) and a comparatively small total mass (approximately 110 g) and is actively involved in vital functions of the cardiovascular system. Two major characteristics of endothelial structure and function have been reported: first its diversity between species, organs, and vessels and secondly, the role of the luminal surface of these cells [23] Crosstalk between cells in the blood vessel wall is essential for vascular homeostasis and the development of diseases. EC dysfunction leads to the expression of adhesion molecules for inflammatory cells, which promotes the development of vascular diseases. The adhesion and migration of inflammatory cells produce oxidative stress, which is a key factor in the initiation and development of vascular diseases.

In contrast, in anaerobic conditions, cardiomyocytes break down glucose to pyruvate via glycolysis and then direct pyruvate into mitochondria to generate ATP via oxidative phosphorylation. The oxidation of pyruvate is regulated by the pyruvate dehydrogenase reaction. Other substrates, including lactate, ketone bodies and AA, can enter mitochondria directly for oxidation.

2.       Quite significant proportion of references are missed in this manuscript. For example, line 42, line 465 the authors discussed the connection of angiogenesis and osteogenesis without any references

3.       More recent works in this field are expected here

In agreement with the Reviewer we modified the paragraph and added new references.

As we reported, this part of our manuscript has been modified in order to focus the aim of the review more informative concerning the connection of angiogenesis and osteogenesis.

P9: Bone is a highly vascularized tissue reliant on the close spatial and temporal connection between blood vessels and bone cells to maintain skeletal integrity. An intricate connection between osteogenesis and angiogenesis exists. Decreasing activity of osteoblasts leads to osteoporosis, and crosstalk between osteogenesis and angiogenesis has been shown to play a vital role in bone regeneration (A.P. Kusumbe, S.K. Ramasamy, R.H. Adams, Coupling of angiogenesis and osteogenesis by a specific vessel subtype in bone, Nature 507 (7492) (2014) 323–328, Osteoblasts Regulate Angiogenesis in Response to Mechanical Unloading. Veeriah V, Paone R, Chatterjee S, Teti A, Capulli M. Calcif Tissue Int. 2018 Nov 21.;). Accumulating evidence supports the role of exosomes secreted EPCs in stimulating angiogenesis, which is closely coupled with osteogenesis (Stem Cell Res Ther. 2019 Jan 11;10(1):12.Exosomes secreted by endothelial progenitor cells accelerate bone regeneration during distraction osteogenesis by stimulating angiogenesis. Jia Y et al   )

P12: Front Cardiovasc Med. 2018 Dec 18;:183..The Interplay of SIRT1 and Wnt Signaling in Vascular Calcification. Bartoli-Leonard F et al

P15: Expert Opin Ther Targets. 2019 Feb;23(2):127-142.. CD73 as a potential opportunity for cancer immunotherapy. Ghalamfarsa G et al  

 4.       Minor point

line 275-278 has a different format with other parts of the manuscript.

Now, as suggested, it is corrected.